# How Good Does This Sound? Examining Listeners' Second Language Proficiency and Their Perception of Category Goodness in Their Native Language

Charlie Nagle [1,*], Melissa M. Baese-Berk [2], Carissa Diantoro [2] and Haeun Kim [3]

1. Department of Spanish and Portuguese, The University of Texas at Austin, Austin, TX 78712, USA
2. Department of Linguistics, University of Oregon, Eugene, OR 97403, USA
3. Department of English, Iowa State University, Ames, IA 50010, USA
* Correspondence: cnagle@austin.utexas.edu

**Abstract:** Language learners often transfer the sounds and prosody of their native language into their second language, but this influence can also flow in the opposite direction, with the second language influencing the first. Among other variables, language proficiency is known to affect the degree and directionality of cross-linguistic influence. However, little is known about how second language learning affects listeners' perception of their native language. To begin addressing this gap, we examined the relationship between learners' second language proficiency and their category goodness ratings in their native language. Thirty-nine English-speaking learners of Spanish listened to English words that began with voiced and voiceless stop consonants and were asked to rate how well the word represented the intended word on a 5-point scale. To create a voicing continuum, we manipulated the voice onset time of the word-initial stop in each target item from 125 ms of prevoicing to 100 ms of aspiration, in 25 ms steps. Proficiency did not affect the perception of voiced targets, but both proficiency and L2 stop production affected the perception of voiceless targets.

**Keywords:** speech perception; category goodness; crosslinguistic influence; second language learning; proficiency; stop consonants; voice onset time

## 1. Introduction

Research on cross-linguistic speech perception and production has shown that the languages a speaker knows influences the other languages that speaker knows. This cross-linguistic influence is bidirectional; the speaker's native language (L1) is known to exert a powerful influence on their perception and production of the second language (L2; e.g., Flege 1987), but the L2 can also have an impact on the L1 (e.g., Chang 2012). This bidirectional influence has been well-documented (e.g., Chang 2013; de Leeuw 2019; de Leeuw et al. 2010, 2012, 2018; Hopp and Schmid 2013; Kartushina et al. 2016; Lang and Davidson 2019; Sancier and Fowler 1997). However, the nature and strength of cross-linguistic interaction appears to depend on several factors, including the age at which the speaker began learning the L2, quantity and quality of L2 input and use, and L2 proficiency. Furthermore, effects may be asymmetrical, as there may be different patterns of crosslinguistic influence for perception and production. Understanding these effects and the variables that regulate them can provide the data needed to refine current models of L2 speech learning. In the following sections, we first provide a targeted review of speech learning models and crosslinguistic interaction in speech production. We then turn to the topic of crosslinguistic interaction in speech perception and the factors that potentially regulate it, which is the focus of our study.

*1.1. Speech Learning Models*

Researchers have proposed several models to account for crosslinguistic effects in speech perception and production and to explain the relationship between the two modalities (Best 1995; Best and Tyler 2007; Escudero 2007; Flege 1995; Flege and Bohn 2021; Kuhl et al. 2008; van Leussen and Escudero 2015). According to Best and Tyler's (2007) L2 Perceptual Assimilation Model, initial crosslinguistic perceptual assimilation patterns set the difficulty of the learning task. If L2 speakers assimilate both members of an L2 contrast to a single L1 category and judge them to be equally good exemplars of that category, then the L2 contrast should be difficult to learn to perceive. If, however, the goodness of fit of the two L2 sounds varies, or if they are assimilated to different L1 categories, then the L2 contrast should be easier to perceive. The L2 Linguistic Perception Model (van Leussen and Escudero 2015) also discusses L2 perception but within an optimality-based approach. Thus, these models focus on explicating L2 perceptual learning, but they do not address how learning an L2 may affect a listener's perception and production of their L1.

One model that does address crosslinguistic interaction in perception and production is the Speech Learning Model (Flege 1995; Flege and Bohn 2021). According to this model, L1 and L2 categories exist in a common space and interact through the mechanisms of phonetic category assimilation and dissimilation. If L2 speakers do not perceive a difference between L2 sounds and perceptually similar L1 categories, then they will not create a new category for the L2 sound. In this case, the L2 will be perceived and produced according to L1 norms, and, given enough L2 experience, the L1 may show the influence of the L2. Conversely, if they do perceive a difference between L2 sounds and perceptually similar L1 categories, then they may create a new category for the L2 sound. This category may, in turn, affect the L1 category, deflecting it away from the L2, such that the contrast between L1 and L2 categories remains robust. In the Speech Learning Model, age of onset of L2 learning (or amount of L2 experience and input relative to the L1, for which age of L2 onset can serve as a reasonable proxy) is hypothesized to be one of the most important regulators of phonetic category formation, insofar as the older the L2 speaker is at the onset of L2 learning, the less likely it is that they will detect crosslinguistic differences. Presumably, however, L2 proficiency and experience could also affect the probability of forming a new L2 category and, by extension, regulate the extent of crosslinguistic interaction.

*1.2. Crosslinguistic Influence in Speech Production*

In examining crosslinguistic influence, researchers typically apply two tests. First, they examine the extent to which bilinguals distinguish between their two languages; that is, whether they show distinct perceptual boundaries and cue weights and if they produce sounds whose phonetic characteristics are significantly different from one another (e.g., Amengual 2011). Second, they compare bilinguals to age-matched monolinguals in perception and production to determine if bilinguals perform like monolinguals in their languages (e.g., Caramazza et al. 1973). Together, this data can shed light on the nature and extent of crosslinguistic influence in bilingual speech perception and production.

With respect to production, accumulated findings indicate that L2 speakers produce a clear distinction between L1 and L2 sounds and produce L2 sounds that are more phonetically accurate as they become proficient in the L2, even though their L2 production rarely aligns with that of a monolingual comparison group (Caramazza et al. 1973; Flege and Eefting 1987a, 1987b). The effect of the L2 on L1 production appears to be more variable. On one hand, some studies have shown that L2 learning does not affect L1 production much at all, as bilingual and monolingual speakers do not significantly differ in their production (Caramazza et al. 1973; Flege and Eefting 1987a, 1987b). On the other hand, in some cases, bilinguals may produce merged phonetic variants that, while still distinguishable from one another, fall between the phonetic norms of the L1 and L2 (Flege 1987). L2-to-L1 effects may even be evident in the earliest stages of L2 learning (Chang 2012), which could be attributable to a novelty effect that catalyzes rapid reorganization and attunement to the L2 (Chang 2013). Crosslinguistic, L2-to-L1 influence is also more likely to occur under certain

conditions. For instance, bilinguals tend to produce cognates whose acoustic characteristics reflect the properties of both their languages (Amengual 2011). Here too, however, these effects are asymmetrical, given that the L1 often exerts a stronger influence on the L2 than vice versa, even in L2-dominant individuals (Antoniou et al. 2011). Individual differences in L1 and L2 input, use, and dominance likely explain some variation in results (Kartushina et al. 2016). It also bears mentioning that crosslinguistic transfer is a dynamic process, such that the effect of the L2 on the L1 and vice versa changes as patterns of input and use change (Sancier and Fowler 1997). Importantly, L2-to-L1 effects in production may also be attributable to perceptual patterns; that is, how speakers map L2 sounds onto L1 categories and how similar they consider those sounds to be.

### 1.3. Crosslinguistic Influence in Speech Perception

A vast body of literature has documented how the L1 affects the initial state, rate, and shape of L2 perceptual learning. Speakers from different L1 backgrounds show distinct perceptual assimilation patterns that influence their ability to discriminate L2 pairs and identify L2 sounds accurately (Flege et al. 1997), and L1 dialectal differences have also been shown to affect the perceptual assimilation of L2 sounds (Chládková and Podlipský 2011). Even speakers from the same L1 background (and potentially the same L1 dialect) show individual differences in perceptual assimilation (Mayr and Escudero 2010) and cue weighting (Kim et al. 2018). Overall, then, accumulated findings point to the important role the L1 plays in conditioning the acoustic and perceptual information to which the speaker attends (Chang 2018). Further, individual differences within the general parameters that the L1 sets can also shape perceptual learning pathways.

The ways in which L1 categories and cue weights affect L2 perception are relatively well understood. However, understanding the configuration of speakers' linguistic systems and the extent to which they interact also entails examining how L2 learning shapes L1 perception. To study crosslinguistic influence in speech perception, researchers have examined phonemic boundaries (Caramazza et al. 1973; Casillas and Simonet 2018; Gorba 2019; Gorba and Cebrián 2021), perceptual cue weights (Dimitrieva 2019), and assimilation patterns and category goodness ratings (Cebrián 2006; Fabra 2005; Flege et al. 1994).

With respect to perceptual boundaries, results suggest diverse (and sometimes contradictory) patterns, including: L2-to-L1 transfer and more L2-like boundaries in learners with more L2 experience (Gorba 2019), no transfer and no evidence of more L2-like boundaries at any level of L2 experience (Gorba and Cebrián 2021), and a merged boundary that represents the properties of both the L1 and L2 in highly experienced bilingual speakers (Caramazza et al. 1973). It should be noted that participant characteristics could account for at least part of the divergence in findings. Participants in Gorba and Cebrián (2021) and Gorba (2019) were L2 learners of varying experience and therefore varying proficiency, but their proficiency would arguably show important qualitative and quantitative differences from participants in Caramazza et al. (1973), who were simultaneous and sequential French–English bilinguals living in Canada.

Perceptual assimilation tasks that require participants to map L2 sounds to L1 categories and rate their goodness of fit can also index crosslinguistic effects. As L2 proficiency increases, speakers may show important changes in the way L2 sounds map onto the L1 and what they accept as a prototypical exemplar of the L1 category. Findings in this area tend to point to no effect of proficiency (Fabra 2005), even when comparing individuals with limited L2 experience to individuals who have spent extensive time living in an L2 environment (Cebrián 2006; Flege et al. 1994). Yet, the way in which proficiency is operationalized matters. When Flege et al. (1994) examined the relationship between participants' pronunciation proficiency and their perceptual similarity ratings, several significant correlations emerged. These findings suggest that pronunciation proficiency may be an important regulator of crosslinguistic influence and that experience, which is variably operationalized as length of residence, years of formal instruction, or a combination of the two, may not be a reliable or appropriate proxy for proficiency.

In summary, there is little doubt that L2 learning affects L1 perception, but the ways in which this effect is realized seem to vary according to study design features and participant characteristics. At present, it remains unclear to what extent proficiency in the L2 moderates the effect of the L2 on the L1. Most studies have examined related, but distinct, correlates of proficiency, such as age of acquisition, length of residence, and amount of previous language instruction, dividing participants into relatively coarse-grained categories of experienced and inexperienced L2 speakers. On one hand, this approach is sensible because the clearest effects should be seen in groups representing different extremes of L2 experience. On the other hand, it prevents a more nuanced understanding of the subtle and potentially gradient ways in which proficiency affects crosslinguistic influence in speech perception.

*1.4. The Current Study*

In this study, we tested whether learning L2 Spanish affected English speakers' perception of stop consonants in their L1. We used a category goodness rating to examine participants' perception of word-initial English stops differing in their voice onset time (VOT). We operationalized proficiency in two ways: as participants' global proficiency in Spanish and as their pronunciation proficiency, which we determined by analyzing their stop consonant production in English and Spanish.

Examining English-speaking learners of Spanish is an interesting test case because of the distribution of contrastive cues across the two languages. In both languages, VOT, a temporal cue that refers to the amount of time that elapses between stop release and the onset of voicing, is the primary cue to stop consonant voicing contrasts. However, as shown in Figure 1 below, phonologically voiced stops in Spanish and English are in a subset–superset relationship (Zampini and Green 2001). In English, phonologically voiced stops may be realized as either prevoiced or voiceless unaspirated stops ([b] or [p]), whereas in Spanish voiced stops are always prevoiced. The same basic observation is true for phonologically voiceless stops, but in the opposite direction. Although in Spanish phonologically voiceless stops are phonetically realized as voiceless unaspirated stops, with VOT values in the 10–30 ms range, aspirated realizations do occur (Rosner et al. 2000). Even if aspirated variants would not be regarded as prototypical exemplars of Spanish voicelessness (Schoonmaker-Gates 2015), they would nevertheless be perceived as phonologically voiceless. As a result of these subset–superset relationships, what listeners consider a prototypical stop is likely to differ between the two languages. Therefore, in the present study, we used prototypicality or category goodness judgments to investigate how learning an L2 (i.e., Spanish) affects perception in the L1 (i.e., English).

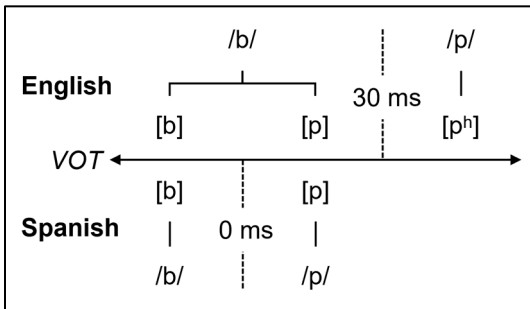

**Figure 1.** Comparison of the phonetics and phonology of English and Spanish stop consonants. The vertical dashed lines represent the approximate crossover points from voiced to voiceless stops.

Our research questions and hypotheses were as follows:

1.  To what extent does L2 proficiency shape participants' perception of L1 category goodness? We predicted that participants who were more proficient in Spanish would perceive prevoiced variants as better exemplars of the English voiced category than individuals who were less proficient in Spanish. For voiceless stops, we predicted that with increasing proficiency, listeners would rate stops with shorter VOT values

as better examples of English voicelessness but that the level of acceptability would diminish at or near the boundary of voicing thresholds in English (~30 ms VOT).

2.  To what extent does this effect depend on place of articulation? We hypothesized that the effect of L2 proficiency on the category goodness ratings would be stronger for bilabial than for coronal stops because bilabial stops share the same place of articulation in both languages whereas coronal stops are alveolar in English but dental in Spanish.

3.  To what extent do participants' L1 and L2 production patterns affect their perception of L1 category goodness? We did not have a strong a priori prediction for this research question beyond an exploratory hypothesis that participants' perception might be aligned with their production, especially if L2 production patterns are conceptualized as a measure of phonological proficiency in the L2 (Flege et al. 1994). Regarding the perception of L1 voiced stops, we reasoned that participants who produced L1 and L2 stops with prevoicing might be more likely to endorse higher ratings for prevoiced variants. With respect to the perception of L1 voiceless stops, we reasoned that participants who produced stops with shorter VOT, particularly in L2, would be more likely to endorse higher ratings for short-lag variants.

## 2. Materials and Methods

### 2.1. Materials

For the perception task, we selected five minimal pairs for English /p/-/b/ and five minimal pairs for English /t/-/d/ (Appendix A). We selected monosyllabic words with the constraint that the words making up each minimal pair were of similar frequency, which we checked by consulting the Corpus of Contemporary American English and the Oxford English Dictionary. We also attempted to select pairs in which the only difference in spelling would be in the stop consonant (save tech-deck).

Target words were recorded by the first author, a male native speaker of American English who is a trained phonetician and a near-native speaker of Spanish. The speaker recorded five tokens of each target word, making sure to prevoice voiced stops. We selected the best token for each word and used it to create a 10-step VOT continuum ranging from −125 ms of prevoicing to 100 ms of aspiration in 25 ms steps. To create the lag portion of the continuum from 0 to 100 ms, we used Winn's (2020) Praat script to manipulate VOT. To create the prevoicing portion of the continuum from −125 to 0 ms, we manually copied and pasted prevoicing in Praat, ensuring that we copied from and pasted into the middle of the prevoiced section of the token at zero crossings. In this way, we extended or shortened the amount of prevoicing to the desired step values. Using the values from the base stimulus described above, we held fundamental frequency at vowel onset constant in all stimuli because it is a secondary cue to stop consonant voicing (Kapnoula et al. 2017; Winn et al. 2013). We then reviewed all tokens, both prevoiced and lag, for naturalness. The final target stimuli set consisted of 100 audio files, ten VOT steps per minimal pair. The speaker also recorded six additional words—cheer, game, knock, rake, shark, wall—ten times, which were included as distractor items.

### 2.2. Participants

Forty participants were recruited online through Prolific (https://www.prolific.co (accessed on 22 January 2023); see Peng et al. (2022) for a discussion of remote data collection and an extensive review of many studies that have used these protocols). Data was collected remotely for two reasons. First, given data collection restrictions due to the COVID-19 pandemic, we were able to safely collect data using remote protocols. Second, this allowed us to draw a more representative sample of participants who have experience with both Spanish and English, especially since we were interested in the role of proficiency. All participants were based in the United States, reported English as their first language, and reported having learned Spanish as an additional language. Participants did not report any hearing or speech problems. One participant was excluded due to poor audio quality on

one of the production tasks. The final participant pool consisted of 19 females and 20 males, with ages ranging from 19 to 40 (*M* = 27.47 years, *SD* = 6.74 years). One participant reported growing up in Puerto Rico, one in Mexico, one in both the United States and Mexico, one in the United States, Canada, and Guatemala, and 35 reported growing up exclusively in the United States. Participants were asked to describe how they learned Spanish, and we used this information to group them. We classified participants who described learning Spanish at home and speaking Spanish with family and friends as at-home learners (*n* = 20), participants who had learned Spanish through classroom instruction and school as instructed learners (*n* = 16), and participants who had learned Spanish through immersion without necessarily having had classroom instruction or at-home exposure as naturalistic learners (*n* = 3). Thus, this participant sample is representative of L2 Spanish speakers from diverse language learning backgrounds. Participants' Spanish proficiency was measured using the MINT Sprint task (Garcia and Gollan 2021), described below.

*2.3. Procedure*

The experiment was administered through Qualtrics (https://qualtrics.com (accessed on 22 January 2023)) and consisted of three parts: a category goodness rating task, a production task, and the MINT Sprint task. In both the production task and the MINT Sprint task, participants were required to record their speech. This was conducted by integrating Qualtrics with Phonic (https://www.phonic.ai (accessed on 22 January 2023)), a web-based research platform that can collect audio recordings. Because the data were collected online, participants recorded themselves using their own devices, and we made no attempt to control the device they used.

On the category goodness task, participants were told that they would be evaluating multiple productions of a set of words to be included in an online spoken dictionary. As such, we did not instruct participants to focus on the word-initial stop. On each response screen, the participants clicked a play button to hear the audio file, saw a prompt indicating the intended word (e.g., "How good of an example of 'pan' is this?"), and used a 5-point rating scale (1 = *bad* and 5 = *good*; only the endpoints were labelled) to evaluate the goodness of the audio file relative to the intended target. Each target item was played with the full VOT continuum, such that, for instance, the entire 10-step /bæn/-/pæn/ continuum was paired with both "How good of an example of 'ban' is this?" and "How good of an example of 'pan' is this?". The 60 distractor items were played twice, yielding 120 total distractor trials. In total, there were 320 trials (20 target items × 10 VOT steps + 60 distractors × 2 repetitions), which were presented in random order. Participants used their mouse to select their rating and were told to take a short break halfway through the experiment.

After the category goodness task, participants were asked to produce words in English and Spanish. The English words were the same 20 target words included in the category goodness task. The Spanish words were eight minimal pairs, four each for /b/-/p/ and /d/-/t/. All words were disyllabic with stress on the first syllable. To derive minimal pairs with a range of vowels that would mirror the properties of the English words, some of the Spanish target words were nonsense words (Appendix A). Words were blocked by language and presented in random order, and the order of language blocks was counterbalanced across participants. Participants produced each word five times and did not receive special instructions on how to pronounce the target words.

Lastly, participants completed the MINT Sprint task in both Spanish and English. The task was blocked by language, and the language order was randomly chosen for each participant. Participants saw the same 80 pictures in each language, divided into an eight-by-ten grid. The task was presented row by row to facilitate scoring. Participants were instructed to name the pictures starting from the top left as quickly and accurately as possible. The Spanish version served as a measure of participants' global Spanish proficiency, and the English version was included to quantify participants' proficiency in Spanish relative to their proficiency in English on a common scale. After completing the MINT Sprint tasks, participants completed a background questionnaire that collected basic

demographic information (age, gender, hearing impairments, etc.) and information on the languages they knew, how they had learned them, and their self-rated proficiency in each of them.

*2.4. Measurement and Scoring*

VOT measurements were taken from the words in the production task. VOT was segmented manually in Praat (Boersma and Weenink 2022), and VOT measurements were extracted using a Praat duration logging script (Crosswhite 2003). Each word was also coded for whether VOT was negative, indicative of prevoicing, or positive, indicative of lag. Figure 2 shows examples of how stop consonants with different VOTs were segmented. The total VOT data set consisted of 7020 possible observations, 453 of which (6.45% of the data) were excluded from analysis because the audio quality was poor, the participant skipped the word, or the participant produced a consonant other than the target stop.

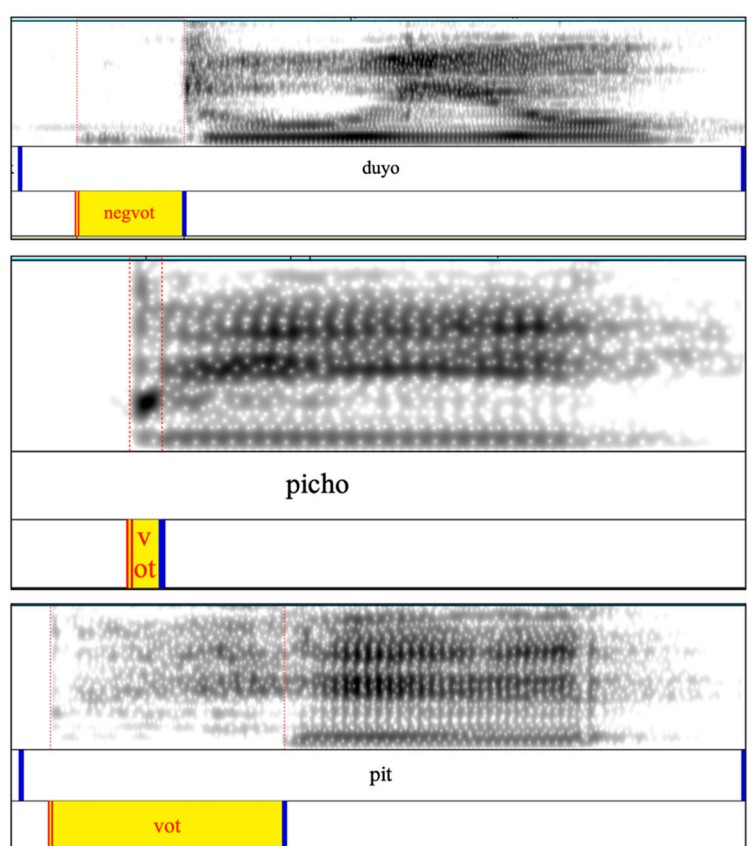

**Figure 2.** Segmentation of a stop consonant with (**Top**) prevoicing or negative VOT, (**Middle**) without aspiration, and (**Bottom**) with aspiration.

The MINT Sprint tasks were scored by two researchers following Garcia and Gollan (2021). In instances where the participant produced a Spanish word that was difficult to identify, the first author reviewed and scored it.

## 3. Results

For this manuscript, we analyzed the production data principally to generate subject-level production estimates to integrate into the L1 category goodness models. For the sake of completeness, we present the full analysis of the production data, but we do not interpret and discuss the production results in detail. Data sets and R code for this manuscript can be accessed at anonymized OSF link.

*3.1. Stop Consonant Production*

　　All data analysis was conducted in R (R Core Team 2022). We used the lme4 package (Bates et al. 2015) to analyze participants' stop consonant production and the lmerTest (Kuznetsova et al. 2017) and sjPlot (Lüdecke 2022) packages to extract model estimates and determine statistical significance.

　　Descriptive statistics are given in Table 1. As shown, there was a 15–30% difference in the mean percentage of prevoicing in English and Spanish, with participants producing more prevoiced tokens in Spanish than in English, especially for /d/. Mean VOT values for voiced stops also reflect a tendency to prevoice /d/ more in Spanish. VOT values for voiceless stops show that participants produced voiceless stops with nearly 55–65 ms less VOT in Spanish than in English on average.

**Table 1.** Descriptive statistics for production and proficiency measures.

| | English | | Spanish | |
|---|---|---|---|---|
| | *M (SD)* | *Range* [1] | *M (SD)* | *Range* [1] |
| VOT /b/ | −61.75 (54.27) | −215–15 | −60.83 (45.27) | −165–13 |
| VOT /p/ | 79.65 (20.15) | 41–123 | 24.71 (15.61) | 10–68 |
| % prevoicing /b/ | 55.91 (33.06) | 0–100 | 69.55 (25.89) | 11–100 |
| % prevoicing /p/ | 0.21 (1.28) | 0–8 | 0.38 (1.77) | 0–10 |
| VOT /d/ | −43.06 (55.45) | −206–34 | −62.38 (39.90) | −142–28 |
| VOT /t/ | 93.32 (21.23) | 57–171 | 30.80 (19.96) | 11–90 |
| % prevoicing /d/ | 45.98 (32.85) | 0–100 | 73.98 (27.50) | 0–100 |
| % prevoicing /t/ | 0.00 (0.00) | na [3] | 0.77 (4.06) | 0–25 |
| Proficiency [2] | 68.56 (6.50) | 45–79 | 46.56 (14.96) | 15–77 |

[1] For VOT, ranges have been rounded to the nearest ms for the sake of presentation. [2] Maximum score of 80. [3] There were no instances of prevoiced /t/ in the English data set.

3.1.1. Voiced Stop Production

　　We fit two sets of models to the voiced data: logistic models in which we analyzed the probability of producing a prevoiced variant and linear models in which we analyzed the amount of prevoicing that participants produced in the tokens they prevoiced. For the logistic model, we report odds ratios, derived by exponentiating the log odds; for the linear model, we report estimates. On the odds ratio scale, ratios > 1 favor the production of prevoicing and ratios < 1 disfavor it. We fit the maximal fixed-effect model of interest, which included fixed effects for Language, Place, and Spanish Proficiency and the Language × Spanish Proficiency interaction term. We then forward-tested by-subject random slopes for Language and Place to determine if those effects varied significantly across subjects. We contrast coded both variables (for Language, English = −0.5 and Spanish = 0.5; for Place, bilabial = −0.5 and alveolar/dental = 0.5; (Linck and Cunnings 2015)), such that the estimate for Language represented the mean difference between Spanish and English and the estimate for Place represented the mean difference between bilabial and alveolar or dental stops. We standardized the proficiency measure.

　　In the logistic model of the voiced data, adding by-subject random slopes for Language and Place significantly improved model fit; for Language, $\chi^2(2) = 134.70$, $p < 0.001$, and for Place, $\chi^2(3) = 8.44$, $p = 0.038$. As shown in the model summary (Table 2), the only statistically significant effect was Language. Based on the contrast coding, where Spanish = 0.5, the odds ratio indicates that participants were about twice as likely ($0.5 \times 4.191 = 2.100$) to prevoice phonologically voiced targets in Spanish on average. It is also worth noting that the odds ratio for the intercept, which approached 2, shows that participants were approximately twice as likely to produce prevoiced variants overall.

**Table 2.** Summary of logistic mixed-effects model fit to the voiced stop data.

| Fixed Effects | Odds Ratio | SE | 95% CI | p |
|---|---|---|---|---|
| Intercept | 1.971 | 0.586 | [1.101, 3.529] | 0.022 |
| Language | 4.191 | 1.420 | [2.157, 8.143] | <0.001 |
| Place | 0.796 | 0.176 | [0.516, 1.228] | 0.302 |
| Proficiency | 0.760 | 0.224 | [0.427, 1.353] | 0.351 |
| Language × Prof. | 1.146 | 0.330 | [0.652, 2.015] | 0.636 |
| **Random Effects** | **SD** | **Correlation** | | |
| By-subject | | | | |
|   Intercepts | 1.719 | | | |
|   Slopes: Language | 1.612 | −0.51 | | |
|   Slopes: Place | 0.496 | −0.08 | 0.16 | |
| By-word | | | | |
|   Intercepts | 0.387 | | | |

      To fit the linear model, we created a subset of the voiced data consisting only of the 1949 tokens (60.28% of the data for voiced stops) in which participants produced prevoicing. This allowed us to examine the phonetic characteristics of prevoicing in each language. As reported in Table 3, participants produced significantly different amounts of prevoicing in Spanish and English, producing less prevoicing in Spanish on average. No other effects reached statistical significance. As in the logistic model, incorporating by-subject random slopes for Language and Place significantly improved model fit; for Language, $\chi^2(2) = 81.28$, $p < 0.001$, and for Place, $\chi^2(3) = 20.69$, $p < 0.001$.

**Table 3.** Summary of linear mixed-effects model fit to the prevoiced stop data.

| Fixed Effects | Estimate | SE | 95% CI | p |
|---|---|---|---|---|
| Intercept | −100.662 | 4.590 | [−109.664, −91.660] | <0.001 |
| Language | 23.809 | 4.817 | [14.362, 33.257] | <0.001 |
| Place | 2.266 | 3.898 | [−5.378, 9.910] | 0.561 |
| Proficiency | −1.281 | 3.703 | [−8.544, 5.982] | 0.729 |
| Language × Prof. | 3.137 | 3.248 | [−3.233, 9.506] | 0.334 |
| **Random Effects** | **SD** | **Correlation** | | |
| By-subject | | | | |
|   Intercepts | 25.765 | | | |
|   Slopes: Language | 18.064 | −0.13 | | |
|   Slopes: Place | 8.140 | −0.70 | 0.32 | |
| By-item | | | | |
|   Intercepts | 7.102 | | | |

### 3.1.2. Voiceless Stop Production

      We fit a linear model to the voiceless data. This model contained the same maximal fixed-effect structure, and we followed the same approach to the random-effect structure, forward-testing by-subject random slopes for Language and Place. The voiceless results should be interpreted with caution because model residuals departed from normality at the tails.

      The model of the voiceless data (Table 4) was similar to the model of the voiced data; by-subject random slopes for Language, $\chi^2(2) = 1101.10$, $p < 0.001$, and Place, $\chi^2(3) = 66.09$, $p < 0.001$, improved model fit significantly, and participants produced a large difference in VOT between the two languages, producing Spanish stops with significantly shorter VOT than English stops. In this model, Place was also significant. The positive coefficient demonstrates that speakers produced slightly shorter VOT in bilabial stops than in coronal stops.

**Table 4.** Summary of linear mixed-effects model fit to the voiceless stop data.

| Fixed Effects | Estimate | SE | 95% CI | p |
|---|---|---|---|---|
| Intercept | 64.157 | 7.565 | [49.325, 78.990] | <0.001 |
| Language | −61.656 | 13.112 | [−87.364, −35.947] | <0.001 |
| Place | 10.064 | 2.894 | [4.390, 15.737] | 0.001 |
| Proficiency | −0.143 | 0.152 | [−0.442, 0.155] | 0.346 |
| Language × Prof. | 0.069 | 0.263 | [−0.446, 0.584] | 0.792 |
| **Random Effects** | **SD** | **Correlation** | | |
| By-subject | | | | |
|   Intercepts | 14.026 | | | |
|   Slopes: Language | 23.876 | −0.13 | | |
|   Slopes: Place | 6.455 | 0.16 | −0.01 | |
| By-word | | | | |
|   Intercepts | 5.600 | | | |

### *3.2. L1 Category Goodness Ratings*

#### 3.2.1. Approach to Analysis

We fit ordinal mixed-effects models to the category goodness ratings data using the *ordinal* package (Christensen 2019). We fit separate models to the data for voiced and voiceless word-initial stops. First, we fit the maximal fixed-effects model, which included the three-way Place × VOT Step × Spanish Proficiency interaction to address the effect of L2 proficiency on participants' ratings and if that effect varied as a function of place of articulation, in line with our first two research questions. We included interactions between VOT Step and participants' VOT production in English and Spanish to examine if their L1 and L2 stop consonant production patterns affected their category goodness ratings, in line with our third research question. We derived the production predictors from the production models by extracting by-subject random effects and computing each participants' model-estimated probability of prevoicing voiced stops and their mean model-estimated VOT production for voiceless stops in each language. All continuous predictors were standardized.

After we fit the maximal fixed effects structure with by-subject and by-word random intercepts, we forward-tested by-subject and by-word random slopes for VOT Step. We used likelihood ratio tests to compare the random effects in stepwise fashion. Once we arrived at the final model, we used the *effects* package (Fox and Weisberg 2018, 2019) to generate and plot model-based predicted probabilities of each response rating for focal predictors.

#### 3.2.2. Voiced Stop Targets

For the voiced stop targets, by-participant random slopes for VOT Step significantly improved model fit, $\chi^2(2) = 512.63$, $p < 0.001$, but the model with by-word random slopes did not converge. We therefore adopted the by-subject random slopes model as the best model of the data. As shown in Table 5, the only effect that reached significance was VOT Step, where the odds ratio below 1.000 demonstrates that participants gave higher ratings to stimuli produced with prevoicing than to stimuli produced with voicing lag. Proficiency was not significantly related to the way participants rated the goodness of stimuli throughout the VOT continuum, nor did participants' L1 or L2 stop consonant production affect their ratings. The fixed-effects portion of the model explained approximately 60% of the variance in the ratings (marginal $R^2 = 0.593$) and nearly 83% considering the full model with random effects (conditional $R^2 = 0.825$).

**Table 5.** Summary of ordinal mixed-effects model fit to the ratings for voiced targets.

| Fixed Effects | Odds Ratio | SE | 95% CI | p |
|---|---|---|---|---|
| Place | 1.551 | 0.558 | [0.766, 3.138] | 0.222 |
| Step | 0.080 | 0.070 | [0.014, 0.451] | 0.004 |
| Prof. | 1.009 | 0.013 | [0.983, 1.036] | 0.492 |
| Step × Prof. | 0.983 | 0.018 | [0.949, 1.019] | 0.350 |
| Place × Step | 1.188 | 0.370 | [0.645, 2.188] | 0.581 |
| Place × Prof. | 0.994 | 0.005 | [0.983, 1.004] | 0.256 |
| Place × Step × Prof. | 1.005 | 0.007 | [0.992, 1.018] | 0.490 |
| *Production covariates* | | | | |
|   Eng. Prevoicing | 1.080 | 0.323 | [0.602, 1.941] | 0.796 |
|   Span. Prevoicing | 0.831 | 0.249 | [0.462, 1.497] | 0.538 |
|   Eng. Prevoicing × Step | 0.774 | 0.317 | [0.347, 1.725] | 0.531 |
|   Span. Prevoicing × Step | 1.272 | 0.522 | [0.569, 2.845] | 0.557 |

| Random Effects | SD | Correlation | | |
|---|---|---|---|---|
| By-subject | | | | |
|   Intercepts | 1.213 | | | |
|   Slopes: Step | 1.659 | −0.94 | | |
| By-word | | | | |
|   Intercepts | 0.397 | | | |

Note. Place was contrast coded (bilabial = −0.5, coronal = 0.5). All continuous predictors—Step, Proficiency, Eng. Prevoicing, and Span. Prevoicing—were standardized.

### 3.2.3. Voiceless Stop Targets

By-subject random slopes for VOT Step, $\chi^2(2) = 582.76$, $p < 0.001$, significantly improved the fit of the voiceless ratings model, and when by-word random slopes were integrated into the model, it converged and showed better fit, $\chi^2(3) = 30.017$, $p < 0.001$. The fixed effects portion of the model explained approximately 72% of the variance in participants' ratings (marginal $R^2 = 0.718$) and the complete model with random effects explained 91% (conditional $R^2 = 0.910$). This model is reported in Table 6, where statistically significant effects are shown in boldface. The fact that the odds ratio for the significant Step × Proficiency interaction term was greater than 1.000 indicates that the more proficient the participant was in Spanish, the more likely they were to endorse higher ratings at steps produced with longer VOT and lower ratings at steps produced with prevoicing. This effect is shown in Figure 3 at ± 2 SD, which represents the extremes of the proficiency measure.

**Table 6.** Summary of ordinal mixed-effects model fit to the ratings for voiceless targets.

| Fixed Effects | Odds Ratio | SE | 95% CI | p |
|---|---|---|---|---|
| Place | 0.991 | 0.363 | [0.483, 2.031] | 0.980 |
| Step | 4.819 | 5.508 | [0.513, 45.263] | 0.169 |
| Prof. | 0.954 | 0.015 | [0.924, 0.985] | 0.004 |
| Step × Prof. | 1.070 | 0.026 | [1.021, 1.121] | 0.005 |
| Place × Step | 0.491 | 0.212 | [0.21, 1.146] | 0.100 |
| Place × Prof. | 0.994 | 0.008 | [0.979, 1.009] | 0.406 |
| Place × Step × Prof. | 1.010 | 0.009 | [0.992, 1.028] | 0.278 |
| *Production covariates* | | | | |
|   Eng. VOT | 1.099 | 0.249 | [0.705, 1.715] | 0.676 |
|   Span. VOT | 0.496 | 0.119 | [0.31, 0.794] | 0.003 |
|   Eng. VOT × Step | 0.899 | 0.303 | [0.465, 1.739] | 0.753 |
|   Span. VOT × Step | 2.433 | 0.860 | [1.217, 4.865] | 0.012 |

**Table 6.** *Cont.*

| Random Effects | SD | Correlation |
|---|---|---|
| By-subject | | |
| Intercepts | 1.285 | |
| Slopes: Step | 1.929 | −0.91 |
| By-word | | |
| Intercepts | 0.224 | |
| Slopes: Step | 0.296 | 0.13 |

Note. Place was contrast coded (bilabial = −0.5, coronal = 0.5). All continuous predictors—Step, Proficiency, Eng. Prevoicing, and Span. Prevoicing—were standardized.

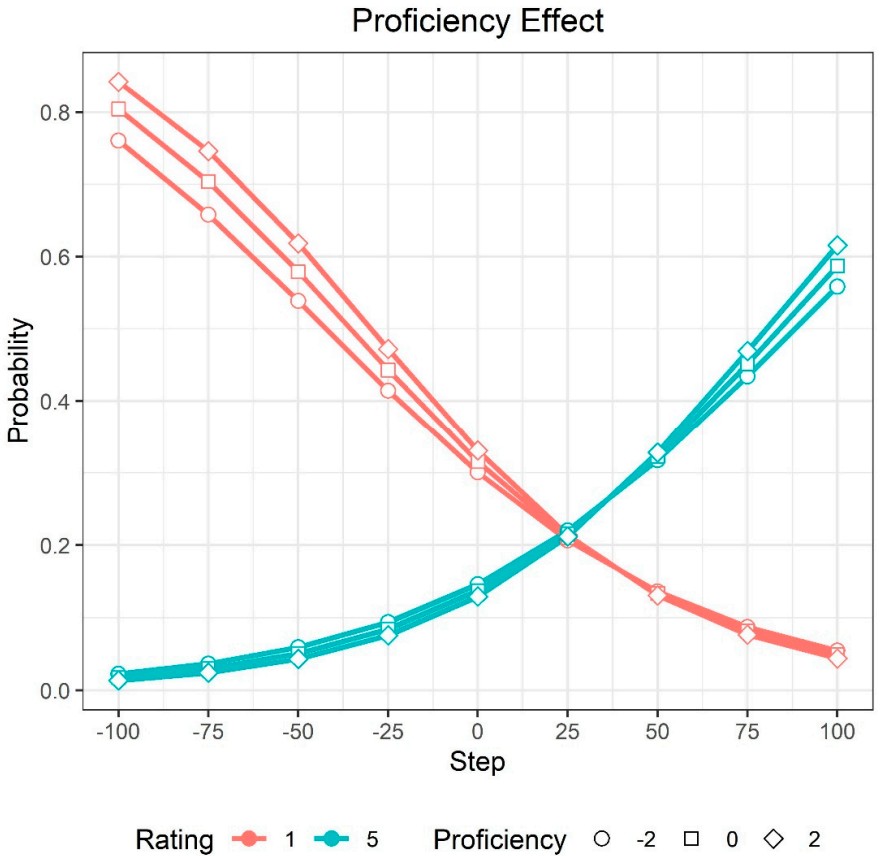

**Figure 3.** Predicted probabilities of response ratings for voiceless trials as a function of participants' proficiency in Spanish. Proficiency was standardized, such that 0 = the group average and ± 2 refers to participants 2 *SD* above and below the mean, which represent the two proficiency extremes in this participant sample. To avoid visual clutter, and because participants tended to choose ratings 1 and 5 much more frequently than the other rating options, only those two probability functions are plotted. On the 5-point scale, 1 = "bad" and 5 = "good".

In this model, participants' production of Spanish stops also had a significant impact on their ratings. As displayed in Figure 4, participants whose Spanish VOT was more English-like tended to show a rating curve that was more closely aligned with English stop consonant voicing boundaries. Furthermore, the more English-like participants' Spanish VOT production was, the sharper the probability curve was, which indicates a more categorical effect for those participants relative to participants whose Spanish VOT was more Spanish-like. For instance, at the 25 ms VOT step, which is near the English stop consonant voicing boundary, participants with very English-like VOT in Spanish (*SD* = 3) had about a 50% chance of endorsing a rating of 1 and a 10% chance of endorsing a rating of 5, with a difference of approximately 40%. Conversely, participants with more Spanish-like VOT in Spanish (*SD* = −1) had about a 25% chance of endorsing a rating of 1 versus a 15%

chance of endorsing a rating of 5, a difference of only 10%. Thus, at the critical boundary, participants with an English-like production strongly rejected voiceless stops produced with shorter-lag VOT, whereas participants with a Spanish-like production were relatively ambivalent, as they were nearly equally likely to endorse either a high or low rating.

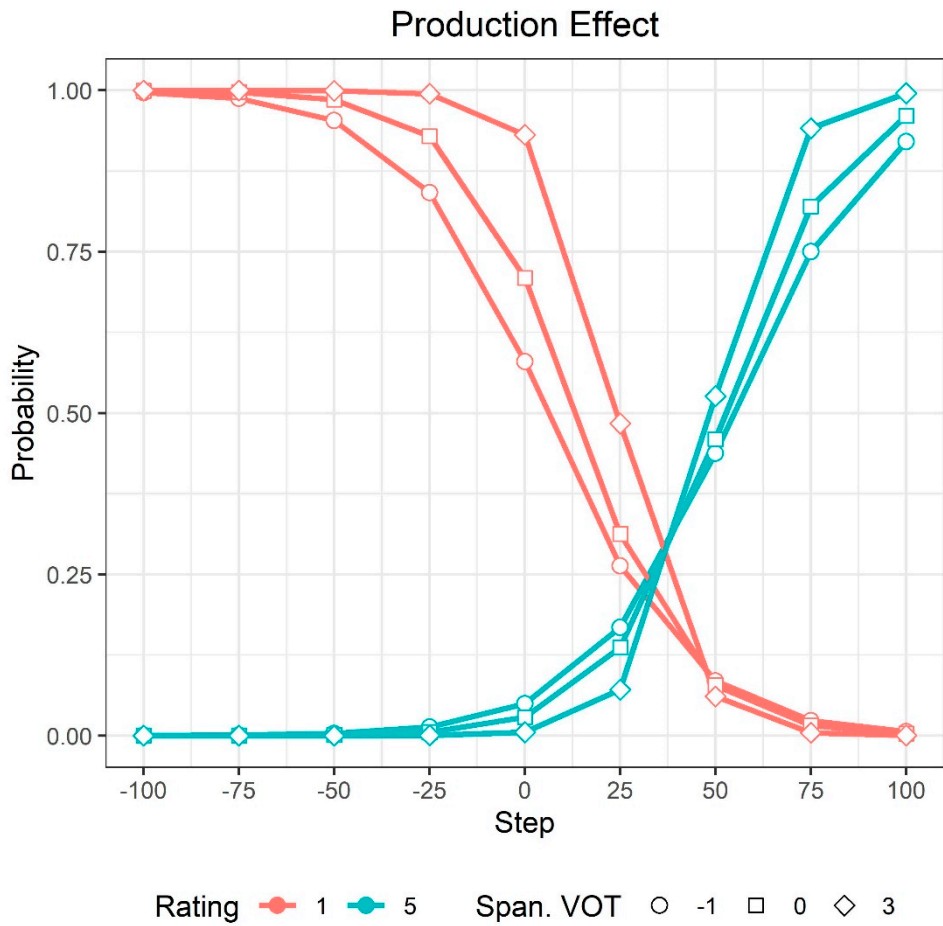

**Figure 4.** Predicted probabilities of response ratings for voiceless trials as a function of participants' model-estimated VOT production in Spanish. Spanish VOT production was standardized, such that 0 = the group average, −1 to a participant 1 *SD* below the mean, and +3 to a participant 3 *SD* above the mean, where −1 and +3 *SD* represent the two extremes of the Spanish VOT production continuum in this participant sample. To avoid visual clutter, and because participants tended to choose ratings 1 and 5 much more frequently than the other rating options, only those two probability functions are plotted. On the 5-point scale, 1 = "bad" and 5 = "good.".

3.2.4. Summary of L1 Rating Results

Overall, proficiency had virtually no impact on how participants rated the voiced targets, which was counter to our initial hypothesis that more proficient speakers of Spanish would rate prevoiced stops as better exemplars of English voiced stops. Further, L1 or L2 VOT production did not appear to affect their ratings. In contrast, for the voiceless targets, both proficiency and L2 production had a significant impact on their ratings. Participants who were more proficient in Spanish were more likely to give higher ratings to stops produced with longer VOT and more likely to give lower ratings to stops produced with prevoicing compared to participants who were less proficient in Spanish. In terms of the effect of L2 production on L1 perception, participants who produced more English-like stops in Spanish—that is, stops with longer VOT—were more likely to prefer stops produced with longer VOT and showed more categorical response curves compared to participants who produced more Spanish-like stops.[1]

## 4. Discussion

### 4.1. Effect of L2 Proficiency on L1 Category Goodness

In this study, we tested whether learning L2 Spanish affected English speakers' perception of stop consonants in their L1. As expected, modeling demonstrated that Step was strongly related to participants' category goodness ratings for word-initial English stops. For voiced stops, participants showed a strong preference for prevoiced variants; for voiceless stops, they showed a strong preference for long-lag variants produced with at least 50 ms VOT. These results show that the category goodness task we implemented worked as intended.

We predicted that with increasing Spanish proficiency, participants would endorse higher ratings for English voiced stops produced with prevoicing, in line with the phonetic norms of Spanish. This prediction was not borne out. For voiceless stops, we predicted that more proficient Spanish speakers would endorse higher ratings for English stops produced with shorter VOT, again in line with the phonetic norms of Spanish. Specifically, we reasoned that they might evaluate the 25 ms VOT step as a better exemplar of phonological voicelessness than lower-proficiency participants. In this case, we found the opposite; the more proficient the participant was, the more they preferred stops produced with the longest lag, and the less they preferred stops with shorter lag, especially voiceless targets paired with prevoicing. Although the magnitude of this effect was relatively small—probability differences between the least and most proficient L2 Spanish speakers rarely exceeded 5–10%—it appears to support an account grounded in the notion of phonetic category dissimilation (Flege 1995; Flege and Bohn 2021), whereby the most proficient L2 speakers seemed to prefer the variants that would create the most separation between L1 and L2 categories. If a prototypically good exemplar of Spanish /p/ is produced with 10–20 ms VOT, then deflecting English /p/ toward higher VOT values, such as 100 ms, would create more acoustic and perhaps perceptual space between English and Spanish voiceless stops. Of course, to fully test this hypothesis, it would be important to collect category goodness ratings in Spanish and to examine the effect of L2 proficiency on the L2 ratings.

An interesting question is why this effect emerged for voiceless but not voiced targets. One potential explanation rests on the subset–superset relationship between English and Spanish stops. In English, the prevoiced variants that correspond to Spanish voicing are phonetically legal, as they are a phonetically acceptable realization of the English voiced category; however, short-lag stops are not an acceptable realization of the English voiceless category. As a result, for voiced stops, it would not be necessary to shift perceptual categories to avoid blurring phonological distinctions and categories across the languages (prevoiced variants are perfectly good exemplars of English voiced stops); however, for voiceless stops, it would, because accepting shorter-lag realizations as good exemplars of English voicelessness would create a fuzzier voicing distinction in the L1.

We also predicted that place of articulation could regulate the strength of the L2 proficiency effect if phonetic similarity drives crosslinguistic interaction. Because coronal stops are alveolar in English but dental in Spanish, we hypothesized that the effect of L2 proficiency would be stronger for bilabial stops and weaker for coronal stops. However, our analysis revealed that place of articulation was not significantly related to participants' perception of category goodness, nor did it interact with L2 proficiency in any way.

### 4.2. Effect of L1 and L2 Production

Although we did not have a strong a priori hypothesis related to the effect of production on L1 stop consonant perception, we reasoned that perception and production might be aligned, such that participants who produced Spanish and English voiced stops with prevoicing would be more likely to endorse higher ratings for prevoiced variants, and participants who produced Spanish stops with shorter VOT would be more likely to endorse higher ratings for short-lag variants. Participants' production in L1 and L2 was not related to their category goodness ratings for English voiced stops, but their production of

L2 stops was related to the ratings they gave English voiceless stop stimuli. Participants who produced L2 Spanish stops with shorter, more Spanish-like VOT showed preferences that were more gradient than participants whose L2 production was more English-like. In fact, model estimates suggest that participants with the most Spanish-like L2 production had a fuzzier sense of category goodness at the critical 25 ms VOT step, the step most closely aligned with the English boundary. At that step, those participants showed a relatively narrow difference in the probability of assigning the highest and lowest goodness ratings relative to participants with a more English-like VOT production in the L2. These findings align with Flege et al. (1994), who reported that participants' pronunciation proficiency, but not their amount of L2 experience, was related to their similarity ratings. Importantly, this finding cannot be attributed to potential collinearity between L2 proficiency and L2 stop consonant production, given that the correlation between the two variables was practically zero ($r = -0.007$). Thus, rather than competing, collinear effects, the proficiency and L2 phonological proficiency—that is, production—effects observed here appear to be additive and independent.

The fact that L1 category goodness was related to participants' own production—albeit their production in the L2 but not the L1—could be taken as evidence of perception-production coordination. However, it is not immediately obvious why participants' Spanish production, but not their English production, would affect their perception of prototypicality in English. Therefore, rather than interpreting these effects as evidence of perception mirroring production, future work must be conducted to clarify the extent to which L1 and L2 perception and production are aligned with one another both within and across modalities.

### 4.3. Limitations and Future Directions

Our finding should be interpreted in light of several important limitations. First and foremost, we did not collect comprehensive language background data related to participants' age of onset of L2 learning and their quantity and quality of L2 experience and use, variables which are known to affect participants' perception and production and the extent of crosslinguistic interaction. Thus, future research should endeavor to collect extensive background information, combining it with measures of global and phonological proficiency, such as the ones used in this study, to determine how these variables relate to one another and affect L2-to-L1 effects. Future work should also aim for larger sample sizes because simultaneously modeling the effect of several participant-level variables will require a sufficiently large sample, and second, larger samples will be beneficial for reliably estimating crosslinguistic effects, which are likely to be modest in size. As a minor note, we also chose to match F0, F1, and F2 values for all stimuli because this information could provide cues to voicing in many languages. Thus, future work could manipulate these secondary cues.

We believe the category goodness task used in this study presents both strengths and limitations. On one hand, by combining scalar judgments with phonetic continua, we were able to observe categorical shifts in perception while at the same time modeling some of the gradience that may be overlooked in perception research. On the other hand, this novel task merits further evaluation. Perhaps using more steps or increasing the number of steps around known crossover points would yield clearer insight into perception within and across category boundaries. For instance, it would have been interesting to examine participants' perception of prototypicality at 10 ms intervals around the Spanish (e.g., $-20$, $-10$, 0 ms) and English (e.g., 10, 20, 30 ms) boundaries. Using finer-grained steps would have also been beneficial for detecting effects associated with place of articulation. Careful consideration of the number of rating options is also important. In the present study, participants made far greater use of options 1 (43.48% of the ratings) and 5 (35.78% of the ratings) than the intermediate options (2 = 5.51%, 3 = 4.81%, and 4 = 10.54% of the ratings). Perhaps this is due to the fact that only the endpoints were labelled, in which case participants may have had a clearer understanding of what 1 and 5 meant in the context

of the research. It could also be the case that five options were simply too many and that a rating paradigm with, for example, three options would be superior. Future research should continue to explore the potential of pairing L1 and L2 category goodness tasks with phonetic continua, taking these recommendations into consideration.

**Author Contributions:** Conceptualization, M.M.B.-B. and C.N.; methodology, M.M.B.-B., C.D., H.K. and C.N.; formal analysis, C.N.; investigation, C.D.; writing—original draft preparation, M.M.B.-B., C.D., H.K. and C.N.; writing—review and editing, M.M.B.-B., C.D., H.K. and C.N.; visualization, C.N.; supervision, M.M.B.-B. and C.N.; project administration, M.M.B.-B. and C.N.; funding acquisition, M.M.B.-B. and C.N. All authors have read and agreed to the published version of the manuscript.

**Funding:** This research was funded by the National Science Foundation, grant number BCS-2117665.

**Institutional Review Board Statement:** The study was conducted according to the guidelines of the Declaration of Helsinki and deemed exempt from review by the Institutional Review Board of the University of Oregon.

**Informed Consent Statement:** Informed consent was obtained from all subjects involved in the study.

**Data Availability Statement:** The anonymized data sets and R code used in this paper can be accessed at https://osf.io/dx9b7 (accessed on 23 January 2023).

**Acknowledgments:** We would like to thank Nick Pandža for his help with the ordinal models reported in this paper.

**Conflicts of Interest:** The authors declare no conflict of interest.

## Appendix A. English and Spanish Target Words

English target minimal pairs for the category goodness task and L1 production task.

| | /p/-/b/ | | /t/-/d/ |
|---|---|---|---|
| pan-ban | /pæn/-/bæn/ | tab-dab | /tæb/-/dæb/ |
| pet-bet | /pɛt/-/bɛt/ | tech-deck | /tɛk/-/dɛk/ |
| pit-bit | /pɪt/-/bɪt/ | tip-dip | /tɪp/-/dɪp/ |
| push-bush | /pʊʃ/-/bʊʃ/ | teal-deal | /til/-/dil/ |
| pair-bear | /pɛr/-/bɛr/ | tuck-duck | /tʌk/-/dʌk/ |

Spanish minimal pairs for the L2 production task.

| | /p/-/b/ | | /t/-/d/ |
|---|---|---|---|
| pala-bala | /pala/-/bala/ | taño-daño | /taɲo/-/daɲo/ |
| peso-beso | /peso/-/beso/ | tela-dela | /tela/-/dela/ |
| picho-bicho | /pitʃo-bitʃo/ | tilo-dilo | /tilo/-/dilo/ |
| puso-buzo | /puso/-/buso/ | tuyo-duyo | /tujo/-/dujo/ |

## Note

[1] Our participant sample included individuals who learned Spanish predominantly through instruction ($n$ = 16) as well as individuals who had learned Spanish at home from family members and could therefore be considered heritage speakers ($n$ = 20). As reviewers pointed out, it is possible that different patterns could emerge for instructed L2 learners and heritage speakers. To evaluate this possibility, we refit the models with a contrast-coded Context of Learning predictor ($-0.5$ = instructed, $0.5$ = heritage) and relevant interaction terms. The associated terms never reached statistical significance in any of the models, and model estimates changed very little as a result of its integration. The full analysis can be accessed in the R code for this paper.

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
