# Peer review of "How Good Does This Sound? Examining Listeners’ Second Language Proficiency and Their Perception of Category Goodness in Their Native Language"

_languages, doi:10.3390/languages8010043_

Round 1

Reviewer 1 Report

This is a very interesting and complete paper that looks at relevant topic in the field of L2 speech and which, in fact, has interesting implications for L2 speech theory. It is also well written and well structured. However, the issue of the effect of L2 proficiency on crosslinguistic perception is not as novel as the study claims and, in fact, relevant references are left out. I do agree that the methodological approach to this issue is very innovative and interesting, but this topic does have a precedence in the literature. I suggest a number of papers that could improve the literature review section. The complete references can be found at the end of this review. I encourage the authors to revise the paper.

Next, more specific comments are provided.

Introduction

P 1 l 26-27. Why listener and not speaker?

p. 1 l 34-35 ‘L2-to-L1 production effects appear to diminish as learners be-34 come more proficient in the L2 (Chang 2012, 2013).’ This applies to beginners who have just moved to the L2 setting, what Chang called ‘novelty effect’. This effect may then diminish and longer stays/greater proficiency may in fact result in greater L2 on L1 influences (check for instance Dmitrieva 2019, Gorba, 2019, Gorba and Cebrian 2021; Sancier and Fowler 1997).  In fact, Gorba 2019 and Gorba and Cebrian 2021 look at L1 and L2 perception of both Spanish learners of English and English learners of Spanish with different degrees of L2 experience.

L .75 ‘to our knowledge, there have not been any studies that have examined the issue of prototypicality, that is, whether  L2 learning shifts listeners’ perception of what counts as a good exemplar of an L1 category, which is what we tested in this study’. à I understand it is not conducted in the same exact manner, but papers on crosslinguistic similarity and L2 on L1 mappings assess a very similar issue, namely the way in which the L2 is perceived through the L1. I suggest you include papers that look at crosslinguistic similarity mappings and ratings, especially those studying the effect of second language experience, for instance, Flege 1991, Flege et al 1994, Cebrian 2006, and Cebrian et al 2019, Rallo Fabra 2005. This has also been assessed from a longitudinal perspective, e.g., Gong, Lecumberri and Cooke (2017).

The introduction points out that there is a research gap when it comes to perception, which is the main focus of the paper. Still, the study also includes data on production, which came to me as a surprise, as no reference to this is made in the introduction or abstract. I believe this should be mentioned earlier in the paper. Moreover, the discussion briefly tackles the perception-production relationship, which has conflicting results in the literature, but this issue is not addressed in the literature review.

Some additional papers on the acquisition of Spanish voiced stops on the part of L1-English speakers include Baese-Berk, 2019; Casillas, 2019; Dmitrieva, et al., 2020 and Nagle, 2019

Materials

Why were those target words chosen? Why specifically those vowels? Is it possible that choosing these vowels – some of them do not seem to have a clear counterpart in Spanish (check Cebrian, 2019 and Escudero and Chladkova, 2010) could have influenced the outcome? That is, perhaps learners were induced into a more English-language mode due to the quality of the vowels and to the lexical meaning attached to the words. Other vowels, such as /i/, have been judged to be crosslinguistically more similar and perhaps non-words – with a phonotactic distribution possible in both languages - would have avoided a possible English-mode.

Were participants instructed on focusing on the initial consonant?

Were secondary cues for voicing controlled in any way?  Some studies point out to an effect of F1 (e.g., Hazan & Boulakia, 1993), especially in those cases were VOT is ambiguous (the crossover area between two categories)

Participants

Were any of the participants heritage speakers? If so, is Spanish really an additional language to these speakers? I think this difference regarding their linguistic background should be addressed somehow, either in the statistical model or in the discussion. Another possibility would be to eliminate these participants from the data set.

Measurement and scoring

VOT values differ according to the vowel that follows, being longer after high vowels. Was context controlled in any way?

I also wonder if the results were completely comparable in the two languages considering the different contexts in which the target phones were embedded. I understand this is a difficult issue to control given the differences between the L1 and L2 vowel systems, but vowel category (or at least a classification in terms of high-low-mid vowels) should somehow be considered in the analysis. It would be nice to at least see the descriptive statistics separately for each context) to see if there were actually differences or not and evaluate whether this should be incorporated in the statistical model.

Discussion

The perception and production link is not addressed in the introduction. I suggest this issue is incorporated, even if just briefly, into the literature review

l. 395-505. No ‘L1 effect whatsoever’ is, I believe, a very strong claim. Dissimilation can be considered in a way an effect on the L1. Even if changes are not in the direction of the L2, category dissimilation does show an interaction between the two systems, i.e.,- a bidirectional interaction between the two languages.

This study is very relevant to L2 speech theory as it addressed key issues of the process of L2 acquisition. These include category formation, crosslinguistic interaction and effects (including effects on the L1) and the relationship between perception and production. The discussion could end with the implications that this study has regarding the L2 speech models, even if, as the authors say, with caution, given its relative small sample size.

Suggested references

Baese-Berk, M. M. (2019). Interactions between speech perception and production during learning of novel phonemic categories. Attention Perception, 81(4), 981. http://mendeley.csuc.cat/fitxers/40f99bf6fdfa610abb22465d179a7f43

Casillas, J. (2019). Phonetic Category Formation is Perceptually Driven During the Early Stages of Adult L2 Development. Language and Speech, 28. https://doi.org/10.1177/0023830919866225

Cebrian, J. (2006). Experience and the use of non-native duration in L2 vowel categorization. Journal of Phonetics, 34(3), 372–387. https://doi.org/10.1016/j.wocn.2005.08.003.

Cebrian, J. (2019). Perceptual assimilation of British English vowels to Spanish monophthongs and diphthongs. The Journal of the Acoustical Society of America, 145, EL52–EL58 https://asa.scitation.org/doi/10.1121/1.5087645.

Cebrian, J., Carlet, A., Gorba, A., & Gavaldà, N. (2019). Perceptual training affects l2 perception but not cross-linguistic similarity. In Proceedings of the 19th International Congress of Phonetic Sciences (pp. 929-933).

Dmitrieva, O. (2019). Transferring perceptual cue-weighting from second language into first language: Cues to voicing in Russian speakers of English. Journal of Phonetics, 73, 128–143. https://doi.org/10.1016/J.WOCN.2018.12.008.

Dmitrieva, O., Llanos, F., Shultz, A. A., & Francis, A. L. (2015). Phonological status, not voice onset time, determines the acoustic realization of onset f0 as a secondary voicing cue in Spanish and English. Journal of Phonetics, 49, 77–95. https://doi.org/ 10.1016/j.wocn.2014.12.005.         

Escudero, P., & Chládková, K. (2010). Spanish listeners’ perception of American and Southern British English vowels. The Journal of the Acoustical Society of America128(5), EL254-EL260.

Flege, J. E. (1991). Age of learning affects the authenticity of voice-onset time (VOT) in stop consonants produced in a second language. The journal of the Acoustical Society of America, 89(1), 395–411. https://doi.org/10.1121/1.400473 ?

Flege, J. E., Munro, M. J., & Fox, R. A. (1994). Auditory and categorical effects on crosslanguage vowel perception. The Journal of the Acoustical Society of America, 95(6), 3623–3641.

Gong, J., Lecumberri, M. L. G., & Cooke, M. (2017). Ab initio perceptual learning of foreign language sounds: Spanish consonant acquisition by Chinese learners. System66, 142-155..

Gorba, C., 2019. Bidirectional influence on L1 Spanish and L2 English stop perception: The role of L2 experience. The Journal of the Acoustical Society of America, 145(6), EL587–EL592. https://doi.org/10.1121/1.5113808.

Gorba, C. &Cebrian, J., 2021. The role of L2 experience in L1 and L2 perception and production of voiceless stops by English learners of Spanish. Journal of Phonetics, 88.

Hazan, V. L. & Boulakia, G., 1993. Perception and production of a voicing contrast by French-English bilinguals. Language and Speech, 36(1), 17–38.

Fabra, L. R. (2005). Predicting ease of acquisition of L2 speech sounds. A perceived dissimilarity test. Vigo International Journal of Applied Linguistics, (2), 75-92.

Nagle, C., 2019. A Longitudinal Study of Voice Onset Time Development in L2 Spanish Stops. Applied Linguistics, 40(1), 86-107, https://doi.org/10.1093/applin/amx011

Sancier, M. L. & Fowler, C. A. (1997). Gestural drift in a bilingual speaker of Brazilian Portuguese and English. Journal of Phonetics, 25(4), 421–436. https://doi.org/10.1006/jpho.1997.0051

Reviewer 2 Report

This study addresses the very interesting question of whether acquiring an L2 influences the perception of phonemic categories in a L1. This manuscript reads well and is, for the most part, easy to follow. A few general changes that I think could improve this paper are to:

1. Spend more time describing 'proficiency' and (especially) 'category goodness' as they are relevant to this study.

2. Provide more detail in the 'Materials and Methods' section (as it is now, future replication of this study would prove difficult).

3. Reconsider the presentation and discussion of the production results…it is fine to only include production as an IV in your analysis of perception, but I do think more explanation and visualisation of your production results (beyond what you have in Table 3) would benefit this article.

4. Redesign Figures 2 and 3, which I found these to be inaccessible and potentially confusing for readers. One thing that might help to do this is to reduce the amount of information shown on each plot, or show different information across several plots.

Please see the attached PDF for other comments and suggestions.

Round 2

Reviewer 1 Report

I would like to congratulate the authors for their work, as the manuscript has been improved considerably and most of my concerns have been addressed. I still have a few comments that should be addressed before accepting the article for publication. You will find the specific comments below.

1.

You should cite some references that support your statements here.

1.1

Last paragraph. Add references at the end of the first sentence.

1.2

First paragraph. Add references

l.140: ‘Spanish-English learners’. Specify the L1 and L2 of the participants in Gorba (2019).

1.4. ‘We also collected data on participants’ production of stop consonants in English as a control measure given that L1 production patterns could correlate with the L1 perception results’ –> I do not fully understand why production was considered a control measure. Could you clarify? In fact, conflicting results have been reported when it comes to links between perception and production (even in the L1).

L. 212. ‘the phonetics and phonology of stop consonants across the two languages’ à perhaps rephrase as ‘distribution of contrastive cues across the two languages under study’ or similar.

2.1. l. 273-274

We held fundamental frequency at vowel onset constant in all stimuli because it is a secondary cue to stop consonant voicing’. What were the F0 values? Were they intermediate between those typically associated to voiced/voiceless stops or were they intermediate?

Moreover, as noted in my previous review, F1 is a secondary cue for voicing. Their potential effect should be acknowledged in the limitations

Reviewer 2 Report

This revised version of this article presents a substantial improvement over the first draft of this article. I commend the authors for taking the time to consider and carefully integrate reviewers' revisions.

In particular, the revisions and additional details in the 'Methods and Materials' section do a lot to improve the clarity and potential replicability of this study. Figure 2 and 3 are an improvement over the figures presented in the first draft of this article. Although they still contain a lot of information, this is not necessarily a bad thing and, indeed, it is probably necessary in this case. I do think that the choice to only represent the 1 and 5 ratings is a wise one.

I like that the discussion is now broken down into different sections, but I would still recommend that the authors lead into this section by first reiterating what the main questions and hypothesis were that they were interested in. This seems important to include, as it means a reader could potentially read only the discussion section and still be able to get the gist of the paper. 

Overall, I think this is a strong paper and it is clear that the authors have worked hard to make it so. Regardless of my suggested change above about how to lead into the discussion section, I think this article is in a polished and publishable state. 
